# METRIC MULTI-DIMENSIONAL SCALING FOR LONGITUDINAL DATA EMBEDDINGS IN PHARMACOMETRICS

## ABSTRACT

Longitudinal data in pharmacometrics typically involves multiple time-varying inputs and outputs for each subject in a population. Each subject can have a different number of observations at different time points, leading to irregular data structures that are difficult to analyze directly. Nonlinear mixed effects (NLME) models are the standard approach for modeling such data, but they can be computationally intensive and may not scale well with large datasets or complex models. In particular, for a large number of input covariates and output biomarkers and endpoints, the computational cost of fitting NLME models can become prohibitive. Some machine learning (ML) methods can be useful in eliminating useless covariates and biomarkers for a relatively low computational budget. Many ML models require fixed-size data as inputs and outputs. Such a tabular representation of a (usually) more complex data structure is commonly known as an embedding. In this work, we generate dissimilarity-preserving embeddings for longitudinal data commonly used in pharmacometrics. We use metric multi-dimensional scaling (MMDS) along with dynamic time warping (DTW) to generate fixed-size embeddings for each time-varying variable of each subject in a population. An experiment on a synthetic pharmacokinetic dataset shows that the proposed procedure can generate useful embeddings that preserve neighborhood structures. This has potential applications in covariate and biomarker elimination as well as model evaluation, to be investigated in future works.

## 1 INTRODUCTION

### 1.1 PHARMACOLOGY AND PHARMACOMETRICS

In pharmacology, drugs administered to the human body are considered to go through the ADME stages: absorption, distribution, metabolism, excretion. Pharmacometrics (Ette & Williams, 2006) is a branch of pharmacology dedicated to modelling these stages quantitatively through mathematical models. The data collected during clinical trials is longitudinal, consisting of time-varying observations for each subject in the study. This data is typically analyzed using ordinary differential equations (ODEs) in nonlinear mixed-effects (NLME) models (Owen & Fiedler-Kelly, 2014) to understand the drug's behavior in the body and its effects over time across a population. The data includes doses given, covariates, drug concentration measurements and various biomarkers that change over time due to the drug's effects.

### 1.2 BIOMARKERS AND COVARIATES

In clinical studies, there is typically a single primary endpoint that serves as the main outcome measured to assess the efficacy of a treatment. However, multiple secondary endpoints and biomarkers are often collected to provide additional insights into the drug's effects and safety profile. Additionally, various covariates are recorded to account for individual differences among subjects that may influence the treatment response. Incorporating these additional variables into the analysis can enhance the predictive performance of pharmacometric models and enable a more precise characterization of drug behavior across diverse patient populations.

The covariates can be either baseline characteristics that do not change over time, such as age, weight, or genetic factors. Or they can be time-varying. On the other hand, biomarkers are typically

dynamic measurements that can vary over time in response to the drug treatment, such as blood pressure, heart rate, or specific protein levels. Often there may be 10s or 100s of biomarkers and covariates collected in a clinical trial, but only a few of them are truly informative for predicting the primary endpoint. Identifying and selecting the most relevant biomarkers and covariates is essential for building accurate pharmacometric models. Machine learning techniques can be used as a preliminary step to identify and eliminate uninformative variables, thereby reducing the final NLME model's complexity.

### 1.3 MOTIVATION FOR EMBEDDINGS

The time-varying variables (covariates or biomarkers) often have a different number of observations per subject across variables and a different number of observations per variable across subjects. This irregular data structure poses challenges for traditional machine learning models that typically require fixed-size tabular data as inputs. While sequence models like recurrent neural networks (RNNs) (Elman, 1990; Werbos, 1990) and transformers (Vaswani et al., 2017) can handle variable-length sequences, they generally require more data and computational resources to train effectively. Given the limited data available in these pharmacometric studies, simpler machine learning models that operate on fixed-size tabular data may be preferred for their efficiency and interpretability.

## 2 METHODS

### 2.1 MULTI-DIMENSIONAL SCALING

Multi-dimensional scaling (MDS) (Borg & Groenen, 2005) is a set of techniques that can be used to create dissimilarity-preserving embeddings for each time-varying variable of each subject in a population. Common uses are dimensionality reduction and data visualization. Metric MDS (MMDS) is a variant of MDS that only requires a pairwise dissimilarity matrix between the points as an input. This makes it suitable for generating embedings from longitudinal data, given a suitable dissimilarity function between subjects' observations. The output of MMDS is a fixed-size embedding for each subject that preserves, as much as possible, the pairwise dissimilarities between the data objects for a given embedding dimension. MMDS formulates the pairwise dissimilarity preservation problem as a non-convex optimization problem, choosing the embeddings that locally minimize the sum of squares of the differences in dissimilarities between the original and embedding spaces. The MMDS formulation has the following definition, where $d$ and $\delta$ are the dissimilarity functions in the input (longitudinal) and embedding spaces, respectively.

$$\min_Y \left( \frac{\sum_{i=1}^{n-1} \sum_{j=i+1}^{n} (d(x_i, x_j) - \delta(y_i, y_j))^2}{\sum_{i=1}^{n-1} \sum_{j=i+1}^{n} d(x_i, x_j)^2} \right)^{\frac{1}{2}}$$

For temporal variables, dynamic time warping (DTW) (Berndt & Clifford, 1994) can be used to compute pairwise dissimilarities between subjects based on their time-varying observations, even if the observation sequences do not have the same length. Notably, DTW is not a proper distance metric since it does not satisfy the triangle inequality. This means that MMDS cannot be guaranteed to find a perfect dissimilarity-preserving embedding. However, in practice, it can still produce useful embeddings that approximately preserve the pairwise dissimilarities.

We implemented MMDS and applied it to a synthetic pharmacokinetic (PK) (Owen & Fiedler-Kelly, 2014) dataset. A simple PK model was built and used to simulate observations for a virtual population of 34 subjects, each with 5 to 15 observations at random time points. More details about the specific model used and data simulated are provided in the supplementary materials. Each subject is represented as a multi-dimensional point defined by their sequence of observations over time. DTW was used to build a pairwise dissimilarity matrix. MMDS was applied to find the optimal embedding of each subject. The limited-memory BFGS optimizer (Nocedal & Wright, 2006) was used for optimizing the embeddings.

## 2.2 EMBEDDING EVALUATION

The embeddings were evaluated using the final MMDS loss objective as well as an experiment measuring how well neighborhood structures were preserved in the embedding space compared to the original space. To better understand the practical utility of the embeddings, we evaluated how well neighborhood structures were preserved in the embedding space compared to the original space. In particular, we found the k-nearest neighbors closest to each subject both in the original input space (using DTW dissimilarities) and in the embedding space (using Euclidean distances). This resulted in 2 sets of k neighbors for each subject. The 2 sets of neighbors were then compared for overlap. Additionally, we calculated the average DTW dissimilarity between each subject and its closest neighbors for each set of neighbors. The average dissimilarity was always calculated in the original space. The specific steps are laid out in Algorithm 1. A value of $k = 5$ was used in the experiment.

---

**Algorithm 1** Embedding evaluation experiment

---

1: Set number of neighbors k
2: Simulate observations from NLME model
3: Determine pairwise dissimilarities of observations using DTW
4: **for** embedding_dimension = 1 to max_dimension_of_embeddings **do**
5:     Initialize embeddings from standard normal distribution
6:     Scale embeddings by average pairwise dissimilarity and embedding dimension
7:     Optimize embeddings
8:     Calculate pairwise distances in embedding space
9:     **for** $n = 1$ to number_of_subjects **do**
10:         Find k closest neighbors in original space
11:         Find k closest neighbors in embedding space
12:         Calculate, in original space, average DTW dissimilarity to embedding neighborhood
13:         Calculate, in original space, average DTW dissimilarity to original neighborhood
14:         **return** Size of intersection of both sets and the average dissimilarities
15:     **end for**
16: **end for**
17: Percentages are $100 * (\text{sizes of intersections})/k$

---

## 3 RESULTS

Figure 1 shows the percentage of subjects that have 4 or 5 overlapping neighbors (out of $k = 5$) in their 2 sets of neighbors as a function of the embedding dimension. As can be seen in the top plot, a considerable overlap between the neighborhoods in both spaces was achieved. This indicates good structure preservation in the embedding procedure.

Figure 2 shows a scatter plot of the average dissimilarity from each subject to its neighbors defined in each space. Each subject is a point with the average dissimilarities representing the coordinates. A 45° is also included as a reference. Points lying on the line represent subjects that are equally distant to both sets of neighbors. As can be seen, good but not perfect agreement across spaces was achieved for embedding dimensions of 3 and beyond.

## 4 CONCLUSION AND FUTURE WORK

We used MMDS and DTW on longitudinal PK data to obtain tabular embeddings. An experiment was performed to validate that the neighborhood structure is preserved in the embedding space. The results indicate that the proposed procedure can generate useful dissimilarity-preserving embeddings for longitudinal data. In the future, we plan to apply the proposed embedding procedure along with tabular ML models to perform covariate and biomarker filtering. The goal is to identify uninformative variables that can be dropped before fitting a final NLME model. Another avenue of research is to use the embeddings for model evaluation by comparing the distributions of embeddings from real data and model simulations, similar to the Frechet Inception Distance used in generative modelling

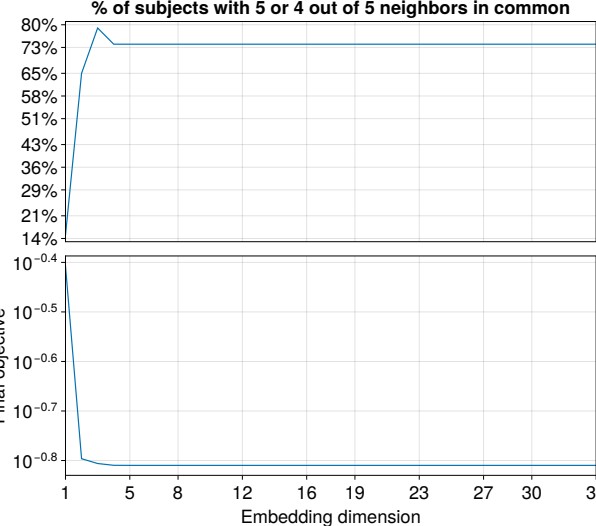

Figure 1: Top plot shows, for each embedding dimensionality, the percentage of overlap between neighborhoods in both spaces. Bottom plot shows final optimization objective values as a function of the embedding dimension.

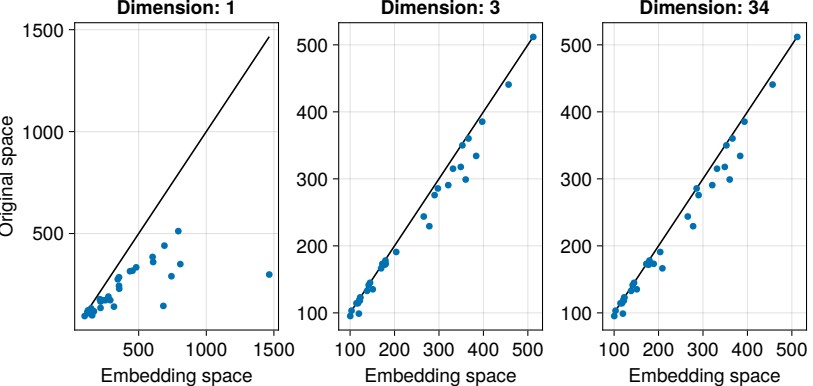

Figure 2: Average DTW dissimilarity to neighbors. Each scatter point represents a subject. The X and Y coordinates are a subject's average DTW dissimilarity to its neighbors in the embedding space and original space, respectively. Proximity to the reference 45° line indicate structural agreement between spaces.

(Heusel et al., 2017). This could provide a new way to assess how well an NLME model captures the underlying data distribution.

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

# A  APPENDIX

## A.1  NONLINEAR MIXED EFFECTS MODELS NOTATION

Let each subject $i \in \{1, 2, \ldots, n\}$ be independently measured at time points $t_{i,1}, t_{i,2}, \ldots, t_{i,m_i}$, leading to the vector of observations $Y = \{Y_{i,j}, \text{for } i \in 1 \ldots n \text{ and } j \in 1 \ldots m_i\}$. For simple Gaussian response variables, the response over time is given by $Y = f(x, \beta) + g(x, \beta, \sigma) \odot \epsilon$, where

- $x$ consists of time $t$ and subject-specific covariates (time-varying or baseline), including baseline covariates $b$ as a component,
- $\beta$ is a vector of estimable parameters, some (random effects) varying across subjects and others not (fixed effects),
- $f(x, \beta)$ is a function whose output has the shape as $Y$ defining the structural model, often defined by a system of ODEs,
- $\epsilon$ is a vector of independently sampled errors of the same shape as $Y$, following a standard normal distribution $\epsilon_{i,j} \sim \mathcal{N}(0, 1)$,
- $\sigma$ is a vector of residual error parameters, shared for all subjects,
- $g(x, \beta, \sigma)$ is a function whose output has the same shape as $Y$ defining the residual error model, and
- $\odot$ is the element-wise product operator.

Let the component of the parameters $\beta$ that is varying across subjects be $\eta_i \sim \mathcal{N}(h(\theta, b_i), \Omega)$ (one vector for each subject $i$), where

- $\theta$ is a vector of estimable parameters, shared for all subjects,
- $b_i$ is the set of baseline covariates for subject $i$,
- $h(\theta, b_i)$ is a function defining the relationship between the baseline covariates for a particular subject and the individual parameters' mean, and
- $\Omega$ is another set of parameters representing the variance-covariance matrix of $\eta_i$, shared for all subjects.

For simplicity, let the remaining part of $\beta$ that is not varying between subjects be part of $\theta$. In other words, $\beta$ can be derived from $\theta$ and $\eta_i$, for all subjects $i$, and the only estimable parameters of the model are:

1. Population-level parameters $\theta$, $\Omega$, and $\sigma$, and

2. Subject-level parameters $\eta_i$ for each subject $i$.

In practice, the PK parameters $\beta$ are often positive-valued so $f(x, \beta)$ will often involve exponentiating the corresponding components of $\beta$. For example, the clearance rate (the rate at which the body metabolizes the drug) of subject $i$ can be defined as $\mathrm{CL}_i = \exp(\eta_{i,1})$ where $\eta_{i,1}$ is the first component of $\eta_i$ with mean $h(\theta, b_i)_1$. It is also common to assume that the $\eta_i$s have a mean of zero, adding their mean value as part of $f(x, \beta)$ instead, e.g.:

$$\eta_i \sim \mathcal{N}(0, \Omega)$$
$$\mathrm{CL}_i = \exp(h(\theta, b_i)_1 + \eta_{i,1}) = \exp(h(\theta, b_i)_1) \cdot \exp(\eta_{i,1})$$

If there are no baseline covariate effects and $h(\theta, b_i)_1 = \theta_1$, this simplifies to $\mathrm{CL}_i = \theta_{\mathrm{CL}} \cdot \exp(\eta_{i,1})$ where $\theta_{\mathrm{CL}} = \exp(\theta_1)$. Since $\theta_1$ is an estimable parameter, we can re-parameterize it as $\theta_{\mathrm{CL}} > 0$ directly. In this case, $\theta_{\mathrm{CL}}$ is the population-level clearance parameter, while $\eta_{i,1}$ represents subject $i$'s log-scale variation from that population value.

The residual error model defined by $g(x, \beta, \sigma)$ can take different forms, e.g.:

1. Additive, where the residual error is constant across all time points and subjects,
$$g(x, \beta, \sigma)_{i,j} = \sigma_1$$

2. Proportional, where the residual error scales with the predicted value $f(x, \beta)$,
$$g(x, \beta, \sigma)_{i,j} = \sigma_1 \cdot f(x, \beta)_{i,j}$$

3. Combined, where both additive and proportional components are present
$$g(x, \beta, \sigma)_{i,j} = \sqrt{\sigma_1^2 + (\sigma_2 \cdot f(x, \beta)_{i,j})^2}$$

### A.2 PK MODEL USED IN SIMULATION

For the experiments presented, PK data was simulated using a simple two-compartment (Central and Peripheral) PK NLME model with combined residual errors. The only covariates in the model were time and the drug's dose. The latent encoding of random effects $\eta$ are considered independent and are sampled from a multivariate normal distribution with zero mean and diagonal covariance matrix $\Omega$ with non-zero elements $[0.25, 0.25, 0.25, 0.25, 0.49]$. The individual parameters are defined below where:

- CL is the clearance rate,
- Vc is the hypothetical volume of the Central compartment,
- $Q$ is the intercompartmental clearance which is the rate of transfer of the drug from the Central to the Peripheral compartment,
- Vp is the hypothetical volume of the Peripheral compartment;
- Ka is the absorption rate constant from Depot to Central.

$$\mathrm{CL}_i = \mathrm{CL} \cdot \exp(\eta_{i,1})$$
$$\mathrm{Vc}_i = \mathrm{Vc} \cdot \exp(\eta_{i,2})$$
$$Q_i = Q \cdot \exp(\eta_{i,3})$$
$$\mathrm{Vp}_i = \mathrm{Vp} \cdot \exp(\eta_{i,4})$$
$$\mathrm{Ka}_i = \mathrm{Ka} \cdot \exp(\eta_{i,5})$$

The vector of population-level parameters $\theta$ is thus $[\mathrm{CL}, \mathrm{Vc}, Q, \mathrm{Vp}, \mathrm{Ka}]$. The specific values of the parameters used in the simulation are included in Table 1.

In the PK model used, the Depot refers to the a compartment associated with the gut for orally-administered drugs. The Central compartment is linked to richly perfused organs and blood plasma. And the Peripheral compartment represents slowly perfused organs, like fat and skin. A compartment name with a prime refers to the rate of change of drug concentration therein.

Table 1: Values of fixed effects $\theta$ used in simulation.

| Parameter in $\theta$ | Value |
|---|---|
| Clearance (CL) | 25 |
| Central volume (Vc) | 50 |
| Intercompartmental clearance (Q) | 31 |
| Peripheral volume (Vp) | 48 |
| Absorption constant (Ka) | 0.5 |

In this simulation, the drug is administered orally so the dose is given into the Depot compartment as a bolus at time 0. The drug amounts (not concentration) over time in each compartment are defined by the following system of ODEs:

$$\text{Depot}' = -\text{Ka} \cdot \text{Depot}$$

$$\text{Central}' = \text{Ka} \cdot \text{Depot} - \frac{\text{CL}}{\text{Vc}} \cdot \text{Central} - \frac{Q}{\text{Vc}} \cdot \text{Central} + \frac{Q}{\text{Vp}} \cdot \text{Peripheral}$$

$$\text{Peripheral}' = \frac{Q}{\text{Vc}} \cdot \text{Central} - \frac{Q}{\text{Vp}} \cdot \text{Peripheral}$$

Lastly, $Y$ represents the drug concentration measurements in the Central compartment. The residual error model $g(x, \beta, \sigma)$ is defined such that a combined residual error model is used. The additive and proportional components of the residual error parameters $\sigma = [\sigma_{\text{add}}, \sigma_{\text{prop}}]$ used in the simulation were both 0.05.

$$\text{Cp} = \frac{\text{Central}}{\text{Vc}}$$

$$Y \sim \mathcal{N}\left(\text{Cp}, (\text{Cp} \cdot \sigma_{\text{prop}})^2 + \sigma_{\text{add}}^2\right)$$

Figure 3 shows the simulated trends of drug concentrations for all subjects.

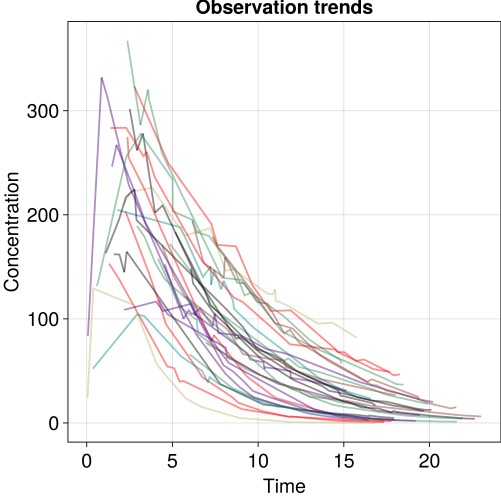

Figure 3: Concentration trends for all subjects in simulated population.

### A.3 LARGE POPULATION

It's difficult to retain the structure across spaces when simulating data with larger populations (here, 75 subjects). This is shown in Figure 4 by the lower percentage of subjects with large intersections of neighbor sets.

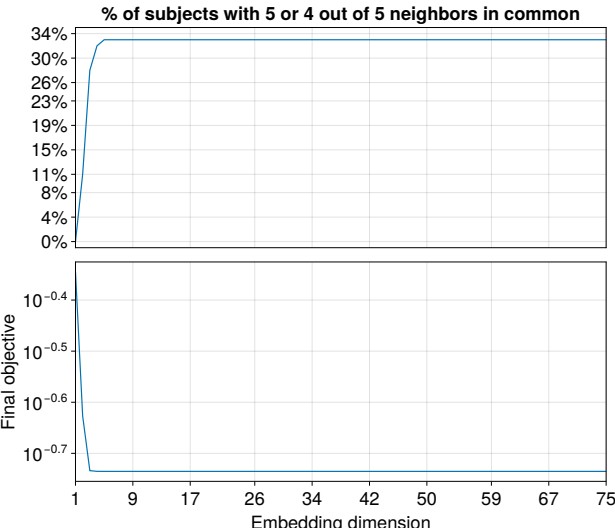

Figure 4: Results of neighborhood overlap experiment with larger population of 75 subjects.

However, as Figure 5 details, the distance experiment still presents good results.

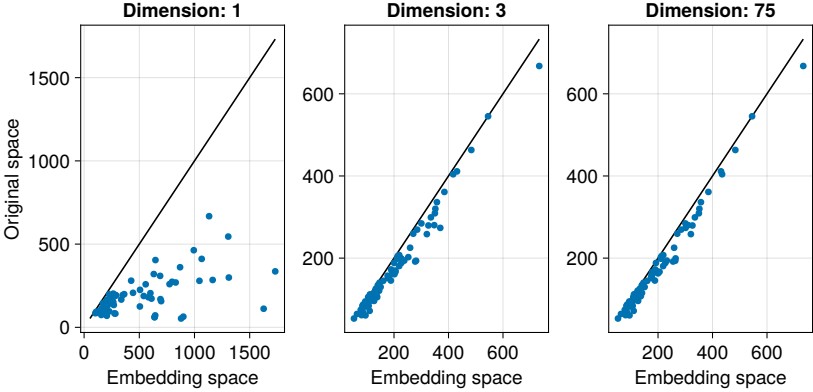

Figure 5: Results of neighborhood distances experiment with larger population of 75 subjects.

