# OpenReview forum: "Metric multi-dimensional scaling for longitudinal data embeddings in pharmacometrics"
_ICLR.cc/2026/Workshop/GRaM — ICLR 2026 Workshop GRaM Poster_

### Official Review · Reviewer_aVZ7 · 2026-02-17
**Metric multidimensional scaling (MMDS) with dynamic time warping (DTW) to generate fixed-size embeddings**

**Rating:** 5
**Confidence:** 3

**Review:**

The authors suggest using metric multidimensional scaling (MMDS) with dynamic time warping (DTW) to generate fixed-size embeddings for irregular longitudinal pharmacometric data. The goal is to preserve dissimilarity structure so that simple tabular ML methods can be applied before fitting expensive NLME models. Experiments on synthetic PK data show reasonable neighborhood preservation in the learned embeddings.

Strengths:

- Simple, interpretable, and solid methodology
- Clear motivation and easy to follow presentation
- I appreciated that the evaluation directly targets geometric structure preservation

Weaknesses:

- Limited methodological novelty (classical DTW + MMDS)
- Experiments are small-scale and synthetic only
- No comparison with alternative embedding baselines
- Practical impact is deferred to future work

The paper is clearly written and easy to follow, and I like the simplicity of the approach. However, the contribution feels somewhat incremental since it mainly builds on existing methods, and the experiments are quite limited, only small synthetic data and no real comparison against other embedding approaches. The idea is reasonable and well motivated, but with the level of validation seems more an exploratory study.

**Pmlr Suitability:**

NA

---

### Official Review · Reviewer_eu36 · 2026-02-24
**A computationally feasible alternative... with no runtime analysis?**

**Rating:** 5
**Confidence:** 3

**Review:**

### Summary:
The paper applies metric MDS with DTW dissimilarities to generate fixed-size embeddings of irregular longitudinal pharmacokinetic data, evaluated on synthetic data from a two-compartment PK model with 34 subjects.

### Strengths:

Clean, well-motivated application of classical methods to an important and real practical problem in pharmacometrics
The neighborhood preservation evaluation (Algorithm 1) is appropriate and clearly described

### Concerns:

The central motivation is computational efficiency relative to NLME model fitting, but no runtime or computational cost analysis is provided. This is the paper's primary justification for existing, and it goes entirely unevaluated. Similarly, the experiment uses 34 subjects with 5-15 observations each. At this scale, NLME models are not computationally prohibitive, which undermines the stated motivation. A convincing evaluation would demonstrate the approach on data where NLME fitting is genuinely expensive, or just a contrast of both methods.

Neighborhood preservation already degrades notably at 75 subjects (Appendix A.3), raising questions about whether the method scales to the regime where it would actually be needed.
I see no comparison to "naive" embedding baselines e.g., functional PCA, signature methods, even simple summary statistics given the nature of the data. This makes it difficult to assess whether MMDS+DTW is the right tool here versus simpler alternatives.

### Overall:
A clearly written exploratory study applying well-established methods to a relevant domain problem. The idea is sound, but the evaluation does not yet support the paper's motivating claims. In particular, the absence of any computational analysis for a computationally-motivated method is a significant gap. The thorough appendix showing negative results was deeply appreciated. Recommending a slight rejection. However, in the spirit of the tiny paper track, I see this as a promising exploratory direction and would encourage the authors to extend the evaluation in a future version.

**Pmlr Suitability:**

NA

---

### Meta-Review · Area_Chair_5BBG · 2026-02-24

**Decision:**

Accept

**Metareview:**

The authors present a metric multi-dimensional scaling for geometric embeddings with applications in pharmacometrics. The reviewers found this paper interesting while having some concerns, and I suggest that the authors incorporate them into the next version of their paper.

**Relevance To Proceedings:**

Tiny paper — does not apply

**Relevance To Workshop:**

Yes — suitable for GRaM

---

### Decision · Program_Chairs · 2026-03-02

Accept (Poster)